# Solving Color Reproducibility between Digital Devices: A Robust Approach of Smartphones Color Management for Chemical (Bio)Sensors

**DOI:** 10.3390/bios12050341

**Published:** 2022-05-17

**Authors:** Pablo Cebrián, Leticia Pérez-Sienes, Isabel Sanz-Vicente, Ángel López-Molinero, Susana de Marcos, Javier Galbán

**Affiliations:** 1Analytical Biosensors Group (GBA), Analytical Chemistry Department, Faculty of Sciences, INMA, University of Zaragoza, 50009 Zaragoza, Spain; cebrianpab@unizar.es (P.C.); isasanz@unizar.es (I.S.-V.); anlopez@unizar.es (Á.L.-M.); smarcos@unizar.es (S.d.M.); 2Complex Systems Group, Polytechnic University of Madrid, ETSI Agronomy, Food and Biosystems, 28040 Madrid, Spain; l.psienes@alumnos.upm.es

**Keywords:** DIC, smartphone, light box, standardization, color reproducibility, RAL, hydrogen peroxide, pH

## Abstract

In the past twelve years, digital image colorimetry (DIC) on smartphones has acquired great importance as an alternative to the most common analytical techniques. This analysis method is based on fast, low-cost, and easily-accessible technology, which can provide quantitative information about an analyte through the color changes of a digital image. Despite the fact that DIC is very widespread, it is not exempt from a series of problems that are not fully resolved yet, such as variability of the measurements between smartphones, image format in which color information is stored, power distribution of the illuminant used for the measurements, among others. This article proposes a methodology for the standardization and correction of these problems using self-developed software, together with the use of a 3D printed light box. This methodology is applied to three different colorimetric analyses using different types and brands of smartphones, proving that comparable measurements between devices can be achieved. As color can be related to many target analytes, establishing this measurement methodology can lead to new control analysis applicable to diverse sectors such as alimentary, industrial, agrarian, or sanitary.

## 1. Introduction

The determination of chemical and biochemical compounds in complex samples is usually based on separation techniques, mainly gas chromatography (GC) or high-pressure liquid chromatography (HPLC), linked to several types of detectors. However, these determinations require fast, low-cost, and easily-accessible methods in many cases. In this case, the methods must be based on portable devices and as smartphones are very widespread, they can become the ideal tool for this task, as long as they can carry out the determinations [1].

According to a study conducted and compiled by two online marketing and communication agencies, We are Social and Hootsuite, in the report Digital 2021 in January 2021 [2], there are about 5.22 billion mobile phone users, which covers 66.6% of the world population. These data show the ubiquity of smartphones and explain why the use of these devices as an analytical tool has grown significantly in the last twelve years [3] (Figure 1). The versatility and ease with which a smartphone is used have led to the development of numerous control analyses: mercury, nitrites, or fluoride in water samples, iron (III) in bioethanol fuel, and formaldehyde in cosmetic products, among many others [4,5,6,7,8,9,10,11,12]. 

All of these methods share the use of the color generated in a chemical or biochemical reaction, as the analytical signal, through digital images, for the determination of the corresponding compound. 

In a quantitative determination, a certain concentration of a dye dispersed or dissolved in a specific support (cellulose sheet, paper strips, etc.) is formed during the development of the method. This concentration corresponds to a specific color, that is, an RGB value. Therefore, any chemical or biochemical colorimetric reaction can be traced with digital still cameras (DSCs).

The ideal situation would be that the information associated with a color was independent of the smartphone used for the measurement. However, as our own experience has suggested in various experiments [13,14], this is never the case. 

In a very simplified way, DSCs integrated into a smartphone take images through three steps: (1) The DSC focuses on the object that wants to be captured through the camera lens, (2) The light enters the lens, passes through a color filter (usually a Bayer mosaic filter, see Appendix A) and the sensor gathers this light through the photodiodes, (3) the hardware of the device processes the light information of the photodiodes, generating and storing a digital image that can be displayed on the screen of the device.

Although this is the general procedure of taking an image by a DSC, color measurements taken in the same lighting conditions are unlikely to be comparable between devices. This is because both filter and sensor are different from one device to another. 

In addition, digital image colorimetry (DIC) is subjected to a series of variables [15,16,17], such as the intensity of the light source used for the measurements, type of illuminant, etc. All these variables make the color measurements not comparable among different mobile devices and induce systematical errors. 

Another problem arises when the hardware of the device transforms the light information into a digital image. It has to transform the sensor gathered information from analogical to digital. This process is carried out using mathematical formulae [18,19] according to the selected image format in which the device stores the image. By default, every DSC can take images in the JPEG format (Joint Photographic Experts Group), which reproduces color in the sRGB (standard Red Green Blue color space), using D_65_ illuminant as reference [20,21]. The mathematical formulae for the sRGB color space associated with the JPEG format are described in Appendix A. Despite the fact that the mathematical transformation for JPEG format is constant to every device, the information gathered by the sensor is not, that is what causes the differences in color measurements between devices. All the images used in this article are taken in this format. Once the image is stored, the device displays the image by illuminating the diodes on the screen of the device.

Furthermore, it is essential to use programs, such as Light Room or Photoshop, to extract the color information from an image [22,23] so it can be treated and represented in any of the color models that exist [24] (see Appendix A). Although these programs are very useful and powerful tools, they focus more on improving the visual quality of the image than on the information it may contain and the use that can be made of it. Other programs or apps, such as Image J or Photometrix, focus on treating the information contained in the image for qualitative/quantitative measurements of color. However, this information still depends on the initial conditions in which the images were taken and cannot be related between devices. That is why all the possibilities of the DIC for color-based control tests have not yet been exploited. 

This article presents a methodology that aims to solve these problems so that the measurements between different smartphones can be comparable. Two strategies are used. The first strategy is to design software using Python^®^ programming language that references all color samples to specific lighting measurement conditions. This software will compare and minimize the differences between the RGB values of a referenced color system (RAL Classic^®^ Color Chart, also specified by the manufacturer in sRGB color space and D_65_ illuminant [25]) and the smartphone RGB measurements. To do so, two data sets in .csv format (a kind of spreadsheet for Microsoft Excel) are needed: one with the RGB values of the RAL Classic^®^ Color Chart specified by the manufacturer and another with its analogs, measured with the smartphone. The program displays the correlation between the data sets and the absolute errors. Since the color values are not exactly the same, the program will generate two matrices to minimize these errors by means of a linear least-squares fit. The first matrix (3 × 3) corrects the chromatic differences between the data sets, while the second matrix (1 × 3) corrects the intensity differences. This is intended to solve the problem caused by the filters and sensors of smartphones.

The second strategy is to develop a light box in which lighting conditions are more homogeneous and adequate for the color measurements. This is achieved by establishing fixed illumination conditions, controlling the intensity/emission geometry of the light source, and the distance between the smartphone and the color sample. 

The RGB values of the RAL Classic^®^ samples will be obtained and analyzed from the photographs taken with each DSC through Image J software. Apart from the RGB color space, the CIE Lab color space will be used to treat and quantify the color measurements [26]. 

The CIE Lab is a color space based on the human visual system, defined by the International Commission on Illumination (abbreviated CIE) in 1976, developed with the intent of creating a standard for color communication so the properties of the color could be quantified and numerical differences between shades could be determined due to the fact that, unlike RGB color space, CIE Lab is device-independent, which means that the coordinates used to specify a color will produce the same color wherever they are applied. To quantify these differences, mathematical equations will be applied in the CIE Lab color space, known as CIE∆E_2000_ (k_L_:k_C_:k_H_) [27], to establish the differences and tolerance limits between the measured colors and the colorimetric reference system. 

As proof of concept, this methodology has been applied to the determination of H_2_O_2_ (test strips enzymatic biosensors) and pH (colorimetric test strips). These two applications have been chosen because they represent the two types of calibration models most frequently obtained using these devices. In the biosensor, the same color with different intensities and polynomial calibration is obtained; in the pH, a change in a color gamut is observed, and an s-shape calibration line. The determinations were made using several smartphones, obtaining comparable results between them.

## 2. Materials and Methods

### 2.1. Digital Imaging Devices

Different devices were used to measure color: HP Scanjet G2410 Flatbed Scanner (Charged-coupled device with a resolution from 12 dpi to 999,999 enhanced dpi at 100 percent scaling) (HP Inc., Palo Alto, CA, USA), Xiaomi Redmi 6A (13 MPx Camera, CMOS-Sony IMX486 Exmor RS sensor, f/2.2 aperture) (Xiaomi Inc., Beijing, China), Xiaomi Redmi 4A (13 MPx Camera, ISOCELL-Samsung S5K3L8 sensor, f/2.2 aperture) (Xiaomi Inc., Beijing, China), Xiaomi Redmi 5 Plus (12 MPx Camera, CMOS-Omnivision OV12A10 sensor, f/2.2 aperture) (Xiaomi Inc., Beijing, China), Xiaomi Mi A2 (12 MPx Camera, CMOS-Sony IMX464 Exmor RS sensor, f/1.5 aperture) (Xiaomi Inc., Beijing, China), Huawei P30 Lite (40 MPx Camera, CMOS BSI-Sony IMX650 Exmor RS sensor, f/1.8 aperture) (Huawei Technologies Co, Guangdong, China), Apple iPhone SE (12 MPx Camera, CMOS BSI-Apple iSight Camera sensor, f/2.2 aperture) (Apple Inc., Cupertino, CA, USA) and Apple iPhone 8 (12 MPx Camera, CMOS BSI-Apple iSight Camera sensor, f/1.8 aperture) (Apple Inc., Cupertino, CA, USA) [28]. 

### 2.2. Digital Imaging Software

The RGB values of the photographic files (JPEG) were analyzed with Image J (LOCI, University of Wisconsin, Madison, WI, USA) [29,30]. (https://imagej.nih.gov, accessed on 28 February 2022). The RGB values were processed and treated with self-developed Python software (version 3.9) in the online platform Project Jupyter [31] (see Appendix A). Factorial analysis and data treatment were performed with Microsoft Excel 2010™ [32].

### 2.3. Light Box

A light box was designed with AutoCAD software and manufactured with a Zortrax M200 3D Printer to measure the color samples. A polylactic acid (PLA) thermoplastic filament (Black Z-PLA filament) was chosen for the structure of the model due to its physical and mechanical properties [33], which made it perfect for portable systems. The use of 3D technology allowed to manufacture and modify the light box at a relatively low cost [34].

Our design is a cubic shape light box with external dimensions of 10 cm × 10 cm × 7 cm and an internal hole of 6 cm × 6 cm × 6 cm. The main piece has an upper part hole where an illuminant, powered with 4 AA batteries, is placed. The smartphone will rest on a complimentary piece working as a lid on the main piece.

The color samples are located on the lower open part of the light box. The focal length between the smartphone and the color sample is 12 cm (Figure 2).

### 2.4. Reagents and Solutions

Britton–Robinson buffer solution (0.04 M borate, 0.04 M phosphate, and 0.04 M acetate, pH 7.0) was prepared from H_3_BO_3_, H_3_PO_4_, and CH_3_COOH and titrated to the desired pH with 2 M NaOH. The different pH solutions were prepared from the Britton–Robinson buffer titrated to the desired pH with 2M NaOH and HCl 2M. All reagents were supplied by Sigma-Aldrich. Hydrogen peroxide stock solution (33% *w*/*v*) was supplied by Panreac (131077.1211). Different concentration solutions were made from this one.

### 2.5. Illuminants

Indirect constant artificial lighting was used for the color measurement: Genie Esaver Bulp by Philips (Amsterdam, The Netherlands) (11 W), Master TL-D Fluorescent by Philips (The Netherlands) (36 W), 6500 K warm white dimmable LED by EGLO (Austria) (7.5 W) and 5050 K white LED by YJHSMT (China) (8 W). Their spectral power distributions (SPD) were characterized in the range of 380–780 nm and measured with an arrangement composed of an optical fiber (Ocean Optics QP600-1-sR) and a compact monochromator (Ocean Optics QE-65,000).

### 2.6. RAL Classic^®^ Reference Colorimetric System

RAL Classic^®^ color chart (was used to standardize the color measurements. This chart is a collection of 213 colors, each of them named with a 4-digit number in combination with the letter “RAL” (it was defined by Deutsches Institut für Gütesicherung und Kennzeichnung; RAL is the acronym of Reichsausschuß für Lieferbedingungen und Gütesicherung The first digit determines the hue of the color (yellow, orange, violet, etc.). The remaining three digits are chosen sequentially. This color chart is referenced in the sRGB color space and D_65_ illuminant conditions. 

For the contrast analysis between the chart and the RGB values obtained with the devices, 96 RAL color samples were characterized and measured initially in the range 380–780 nm with a spectrophotometer CM-2600d (Konica Minolta, Osaka, Japan). The spectrophotometer uses pulsed Xenon lamps to illuminate the samples and gathers their reflectance spectra. Then this information is transformed mathematically into the desired color space and can be compared with the measurements of the device. The 96 color samples from the chart were chosen so a homogeneous distribution of all types of colors could be introduced to the studies. This was achieved by randomly selecting RAL samples with different hues whose R, G, or B coordinates were different from 0 or 255. As the RGB system range goes from 0 to 255, a value of 0 implies that the sensor has not captured enough light to generate a signal in any of the RGB channels (underexposure) and a value of 255 implies that the sensor has captured too much light (overexposure), with its respective irreversible information loss.

A USB Dino-Lite AM2111 microscope was used to evaluate the integrity of the color samples’ surfaces. A defect in the color of the surface causes an erroneous measurement because the light does not reflect homogenously. To evaluate the integrity of the surfaces, each color sample was inspected individually using the USB microscope. If the surface showed any kind of defect, the sample was considered as “unsuitable” and was discarded. The microscope was placed directly on the surface of each color sample and visualized using the Dino Capture 2.0 program, adjusting the focal length with the microscope to 60 mm.

### 2.7. Procedure

To evaluate the DSC’s initial RGB values and apply the correction method, several steps are necessary. The first step corresponds to the initial evaluation of the color samples and it is where more time is spent within the methodology (about an hour) as the images of the RAL chart have to be taken individually with the smartphone, measured, and filed on a CSV spreadsheet. Once this step has been completed, this part does not have to be repeated anymore (i.e., every time the smartphone is used for a color measurement). The rest of the procedure is very fast as they are carried out with the program that we have designed in Python (Steps 2 and 3, about 5 min overall) and the application in real samples is also implemented in an Excel macro (Step 4, less than 1 min for each measurement). All the steps are outlined in Figure 3.

#### 2.7.1. Initial Evaluation of Color Samples in Digital Images (Step 1)

Firstly, JPEG format photographs were taken with the different devices from each individual color sample. The photographs were imported to the Image J program and a 500 kpixel area was selected to obtain the mean of the RGB values (named as R_0_, G_0_, and B_0_, respectively) with their standard deviation. Then, RGB values were imported to Excel for further analysis. To evaluate the initial differences between the photography color samples and their references, RGB values must be transformed into the CIE Lab color space through a series of mathematical equations [18,26,35] (see Appendix A).

To quantify the differences between the CIE Lab measured color values and their references, a series of mathematical formulae called CIE∆E_2000_ (k_L_:k_C_:k_H_) is used, being k_L_, k_C_, and k_H_ correction magnitudes associated with the observation conditions of the sample (see Appendix A). If each color, measured and referenced, is placed in the CIE Lab 3D color space, this mathematical formula calculates the Euclidian distance between these two colors. This distance is called ∆E. Low ∆E values indicate greater accuracy between the displayed color and the original color standard of the input content, while high ∆E values indicate a significant mismatch. 

For this work, CIE∆E_2000_ (2:1:1) is used, in contrast to what is usually used, which is CIE∆E_2000_ (1:1:1), as it has a better performance when it comes to color comparison in controlled lighting essays [36,37]. 

As a general guide, ∆E values greater than 5 are considered unacceptable in most processes since they indicate that the color difference is especially evident. If ∆E approaches 2.3, we would be talking about the JND or “just noticeable difference,” a very hardly noticeable difference between colors used as a criterion for evaluating color differences acceptability [38]. As the screening ∆E value is dictated by the development of different industries in which it is used, for the first experiments in this work, the JND criteria are chosen and all color samples with an ∆E ≥ 2.3 are considered as inaccurate and are discarded. 

#### 2.7.2. Correction Matrices Generation (Step 2)

The second part of the procedure is focused on the correction of the color differences between the DSCs and reference color samples. To correct these differences, a code written in Python^®^, based on a linear fit using least squares, was used. This code generates two matrices through the comparison of the device’s measured color values and the standards so that the differences are minimized. A general equation of this correction (Equation (1)) is expressed as follows, being [RGB]_0_ the initial color values and [RGB]_c_ the corrected color values:[RGB]_c_ = [M_1_]·[RGB]_0_ + [M_2_](1)

#### 2.7.3. Color Calibration (Step 3)

The next step is to evaluate whether the methodology used has worked. The initial color values are corrected with the Python-generated matrices, transformed again to the CIE Lab system, and the ∆E values are recalculated. If the ∆E mean of all color samples is lower than 2.3, the correction method and the matrices generated can be applied to real samples.

#### 2.7.4. Application of the Methodology and Matrices in Real Samples (Step 4)

Finally, the methodology and the matrices generated previously will be applied to real samples to correct and reference these samples to specific lighting conditions. To do this, a photograph of the color involved in a chemical or biochemical reaction is taken in the same conditions as the RAL Classic^®^ Color samples. Once the initial RGB values are imported to Excel, through all the processes already mentioned, the correction matrices are applied, and the color measurements can be compared between different smartphones. 

## 3. Results and Discussion

### 3.1. Characterization of the RAL Classic^®^ Color Chart Using a CM-2600d Spectrophotometer (Konica Minolta)

The RAL Classic^®^ Color Chart will be used as a reference (96 samples, whose color coordinates are provided by the commercial company). The first study was to characterize them to ensure that they could be used as a reference. For this, we used a Spectrophotometer (Konica Minolta, CM-2600d) (see Section 2.6) and the CIE Lab color sample values obtained were then compared with the references. The comparison showed that all 96 samples had ∆E < 5, but 15 of them did not meet the criteria of JND (∆E < 2.3), as it is shown in Appendix A. Using a USB Dino-Lite AM2111 microscope (see Section 2.7), imperfections or deterioration in their surface were found (due to the use of the color chart) (see Appendix A), so they were discarded, having 81 color samples for the next study. 

### 3.2. Correction Method for Images Taken with an HP Scanjet G2410 Flatbed Scanner

An HP ScanJet G2410 flatbed scanner was first considered as a starting point to evaluate the possibilities of the correction method in a digital system as it was constructed to have controlled lighting conditions. This is achieved by: fixed measurement distance and angle, constant background, and fixed illuminance. 

First, a scan of the 81 RAL color samples was performed to evaluate the differences between the RGB values of the device and the referenced ones. All automatic correction functions associated with the software bundled with the scanner were disabled to ensure that the RGB values of the samples were not manipulated. Once the images were taken, all of those color samples whose R, G, or B coordinates were equal to 0 or 255 were discarded (see Appendix A). Later, each RAL color sample was analyzed and evaluated individually with CIE∆E_2000_. The results were given using the average values of L, a, b, and ∆E_2000_ belonging to the 55 RAL (Table 1 Without Correction). As it can be seen, ∆E2000 has a value of 6.88 ± 0.45, which does not meet the JND standard.

After this first evaluation, the RGB values of the device of all samples were introduced in the Python software and corrected. Then, the new RGB values were transformed into CIE LAB and evaluated again with CIE∆E2000 (Table 1 With Correction), so the results with and without the correction could be compared. 

As it can be seen, the application of the proposed correction method makes ∆E2000 less than 2.3 for the average of the samples (∆E2000 = 1.95 ± 0.24). This shows that the color coordinates measured with the digital device correspond to the referenced ones and the influence of the sensor and lighting on the measurement could be corrected.

The 55 samples that remained will be used to implement the methodology on smartphones, which is the next step.

### 3.3. Study of the Lighting Effect on Colors of Digital Images Taken with a Smartphone

One of the main problems in reproducing color is the lighting conditions under which the color sample is measured [39]. The illuminant used to measure a color determines the RGB values that the photography has, so when a digital image is taken and whenever possible, the lighting conditions must be specified to reproduce color on another device. Each light source has a different spectral power distribution, affecting the amount of light that a color sample will absorb and reflect. Another problem is generated by the directional differences in the color sample. This is caused by the relative position between the illuminant, the sample, and the instrument, causing color variations on the surface of a sample.

In the previous study, this problem did not exist as the digital images were taken under controlled lighting conditions, but that is not the situation in which one works with a Smartphone. 

In this study, we will address the effect of different types of illuminants on color reproduction and how they affect the generation of the correction matrices. Figure 4 shows the differences in the spectral power distribution of all illuminants compared to the referenced illuminant, D_50_. 

The photographs of the RAL Classic^®^ Color Chart samples were taken using a Xiaomi Redmi 6A Smartphone and subsequently treated. Table 2 shows the different results obtained before and after the correction method. The device was placed directly above the sample at a distance of 12 cm and 50 cm from the illuminant. 

The results in Table 2 show that despite the spectral power distribution differences of the illuminants, the correction method can be applied to any kind of illuminants, achieving better results with those assumed to have a spectral power distribution more similar to the D_50_ illuminant in our case, the “YJHSMY White LED Strip.” A complementary study was made to prove these similarities, the Spectral Similarity Index method (SSI) [40,41] (see Appendix A). Based on the results, the “YJHSMY White LED Strip” illuminant will be used for the rest of the experiments in this article.

It is observed that, although it improves in all cases, ∆E2000 is more major than 2.3, probably due to the fact that there is no controlled lighting environment, which is what is studied in the following section.

### 3.4. Controlled Lighting Conditions with a Light Box

Despite the good results in the previous study with the “YJHSMY White LED Strip” Illuminant, all the color samples presented relatively high standard deviation values (SD) in each RGB channel compared to the ones obtained with the flatbed scanner (Table 3). These differences are associated with the fact that the flatbed scanner measurements are taken under fixed lighting conditions due to a light box structure, avoiding the influence of external illuminants and light gradients. 

To verify these problems, a light box was designed and manufactured (see Section 2.3) with a 3D printer so that it could be adapted to smartphones. Three studies were made: (1) a graphical representation of each RGB channel individually, or histogram, to evaluate the differences in the results by means of a light box; (2) a 3D surface plot of a selected area in the image, to evaluate light gradients and, (3) the effect of external illumination inside the light box (see Appendix A). Once confirmed that the use of a light box eliminates the influence of external lighting on the measurements and that the light gradients can be corrected, a new SD study of the RGB channels was made. All the studies were performed using the Image J software, comparing the Xiaomi Redmi 6A results before and after using the light box (Table 4). As it can be seen, the standard deviation values have been considerably reduced in all channels.

These new results obtained with the Xiaomi Redmi 6A are consistent with the ones obtained with the HP Scanjet G2410. The use of a light box minimizes the variability of the measurements, achieving more precise color measurements. 

The light box results were also compared using the CIE∆E_2000,_ as shown in Table 5, demonstrating once again the need to carry out color measurements under fixed lighting conditions. As it can be seen, the value of ∆E_2000_ that was obtained in Table 2, with the YJHSMY White LED Strip illuminant (2.61 ± 0.27), has been reduced to (2.01 ± 0.22) using the light box.

The rest of the studies in this article will be performed using the light box.

### 3.5. Study of Color Reproducibility between Devices. Qualitative Method

The last problem to be solved in color measurements is related to the sensors among smartphones. Each device is manufactured under the specifications of the commercial house, which means that the sensor it uses is unique, making color measurements between devices not comparable due to differences in light gathering.

The objective of this study is to check if both corrections and methodology used so far decrease the variability of the measurements between devices, making them comparable. If this is the case, this methodology will be applied to colorimetric reactions for control analysis.

The first step was to generate the matrices of each individual device, using the 55 RAL and the light box, and apply the correction method. The CIE∆E_2000_ values were evaluated before and after the correction through the photographs taken by each device (see Appendix A). The results showed that the initial measurements of color are not comparable between devices, which is consistent with the use of different sensors. 

Once the corrections were evaluated, 10 RAL samples were chosen unrelated to those used for the correction method. After taking the corresponding photographs with each device, the colors were analyzed and corrected with the matrices. Once corrected, they were compared to each other and with the referenced RGB values before and after the correction method (Table 6). This table shows the RAL value provided by the manufacturer of the RAL chart (Reference) as well as the RAL value obtained without correction (WOC) and once the correction has been applied (WC). Match (Yes/No) means if there is an agreement with the reference value. The results were implemented with a macro developed on an Excel Spreadsheet.

As it can be seen, before applying the correction, there is only correspondence with two RAL (4011 using Xiaomi Redmi 4A and 8025 using iPhone SE). Once the correction is made, there is a correspondence of 4 RAL (2008, 4011, 6010, and 8025) in the three devices. Despite the fact that the rest of the color samples do not correspond with the color chart, they present it between devices (1018, 3016, 5007, 7001, and 9017) except the RAL 9010. 

This is because some colors on the chart have very similar RGB values and the color measurements have certain errors associated with them; therefore, the concordance between colors is not perfect.

Finally, 5 samples from the 10 chosen for the study were randomly chosen (Samples 1, 3, 6, 7, and 9), and the RGB values of each device were represented on a box and whisker plot, comparing these values before and after the correction so that the variability of the measurements could be noticed (Figure 5a–c). This process will allow evaluation of the validity of the correction method and, if it is feasible, to apply it to real color samples.

Since the methodology has been proven to work and comparable measurements between devices can be achieved, the next step is to focus on chemical colorimetric reactions for control analysis.

Two applications are studied: in the first one, there are changes in the intensity of the same color for different concentrations of analyte (H_2_O_2_), and in the second, there are different colors for different analyte concentrations (H^+^).

### 3.6. Measurement of H_2_O_2_ in Various Solutions Using 3,3′,5,5′-Tetramethylbenzydine (TMB) as the Reaction Colorant. Quantitative/Qualitative Method on Test Strips

The first study was centered on the determination of H_2_O_2_ in various solutions. Numerous brands of peroxide test strips are commercially available. For this study, Quantofix Peroxide 25^®^ test strips will be used. These strips contain a Horseradish Peroxidase-like enzyme to catalyze the reaction and 3,3′,5,5′-Tetramethylbenzydine, which acts as the colorant of the redox reaction. (see Appendix A). 

The color changes produced on the strips occur between 0 to 25 mg/L of H_2_O_2_, generating the graduation of a cyan-like color. 

For the determination of peroxides in these strips, 12 solutions of known concentration were prepared and measured to generate a calibration plot, representing R_0_-R/R_0_ vs. the peroxide concentration values (Figure 6). The R coordinate of the RGB system was chosen as an indicator of the color differences between measures due to the capacity of the colorant to absorb at the wavelength of 650 nm. As was previously demonstrated in previous papers [13,14], a second-order degree polynomial relationship between (R_0_-R)/R_0_ and the concentration of the chromophore is theoretically expected.

Each solution was injected onto the strips and measured after 15 s with the Xiaomi Redmi6A Smartphone, as specified in the strip instructions. Then the RGB values were corrected and used for the calibration. 

Then, four peroxide solutions of unknown concentration were prepared to interpolate into the Xiaomi Redmi 6A calibration plot applying the corresponding correction with different smartphones, Table 7. Each solution was measured three times with each smartphone. All the procedure was carried out with a macro developed in Microsoft Excel, as in the previous study, so that the value of the corrected concentration was displayed. At present, a mobile application is being developed to implement all of this methodology into several colorimetric reactions to make this kind of measurement easier.

This study proved that a single calibration with a smartphone can be related to other devices’ measurements and that these measurements can be interpolated in the same graph obtaining good results. It also showed that in all cases, the correction method decreases the relative errors significantly and the variability between measurements (see Appendix A), which corroborated the importance of using this method. Sample 4 was the only sample in which the initial interpolation could not be applicable, but this problem was solved after the correction. For the rest of the measurements, the relative errors are comparable to those of a semi-quantitative method. Previous studies of this reaction in the laboratory [14] showed that the color values of the measurements are dependent on time, so the stabilization of the color in this reaction will be considered in further studies to improve these results. 

### 3.7. Determination of the Different pH Colors of Various Solutions Using Universal Indicator Test Strips

The second study was centered on the determination of the pH color of various solutions. For the analytical application, the pH strips of the PanReac AppliChem^®^ brand were chosen. These strips change their color depending on the pH of the solution due to the variety of indicators they are made of, typically a mixture of thymol blue, methyl orange, methyl red, bromothymol blue, and phenolphthalein [42]. 

First, 12 different solutions in the range of 2 to 14 in pH were prepared from a Britton-Robinson buffer so their colorimetric reaction with the pH strip could be used as a reference for the determination. The strips were immersed for one second in each solution and after 15 s, their color was measured. The generated colors were measured with a Xiaomi Redmi 6A smartphone. The RGB values of the different pH solutions are shown in Table 8. 

These measurements showed that there was not an evident dependence of any RGB coordinate with the pH values (see Appendix A), so another color space, the CIE xy, was considered for the determination. This 2D space is related to the predominant wavelengths of color, can easily be related to the RGB color space through a series of mathematical formulas (see Appendix A), and, as there is a wide color gamut, it is perfect for this kind of determinations. From the two coordinates of this color space, the x coordinate was chosen because it could be related to the pH values by means of a logistic function, as seen in Figure 7. For this article, the Xiaomi Redmi 6A pH calibration will be used to compare the quantitative measurements of three different devices.

To do so, first, the logistic function was established (Equation (2)), which can be expressed in a general way as
(2)$$y=A+\frac{B-A}{1+{\left(\frac{\mathrm{pH}}{C}\right)}^{D}}$$

This function can be linearized as follows:(3)$$y=A+\frac{B-A}{1+{\left(\frac{x}{C}\right)}^{D}}\to \frac{B-A}{y-A}-1={\left(\frac{\mathrm{pH}}{C}\right)}^{D}\to \log\left(\frac{B-A}{y-A}-1\right)=D\log\mathrm{pH}-D\log C$$

Once the logistic and the linearized form functions of the measurements were obtained (Figure 8a,b), 4 different buffer solutions were selected from the laboratory to determine their pH value. Each buffer was measured 3 times with 3 different smartphones. The study results (Table 9) were obtained by interpolating the x value of the CIE xy system of each sample in the linearized form of the logistic function. 

This study proved once more that a single calibration with a smartphone can be related to other devices’ measurements and that these measurements can be interpolated in the same graph, obtaining precise and with low uncertainty results, even better than in the peroxide determination. It also showed that the relative errors decreased significantly after the correction method in all cases, achieving values lower than 2.3% in samples 1, 2, and 3, which are comparable to a quantitative method. The relative error of sample 4, lower than 16%, is consistent with the area in which the sample is interpolated on the logistic function, which has no variation of the x coordinate despite the pH. The variability of the measurements between samples also decreased (see Appendix A), except for sample 4, which could not be initially interpolated on the graph.

## 4. Conclusions

The use of smartphones for colorimetric detection has become the main focus of low-cost analytical measurements. This article proposed a color correction method to make measurements between smartphones comparable in such a way that when future colorimetric studies are developed, the results obtained are reliable and independent of the type of smartphone used.

To achieve comparable colorimetric measurements between devices, a calibration pattern is needed to correct the variance that generates the sensor of the camera. It is also important to characterize the illuminant used in this type of study to know which wavelengths are predominant and what can be expected from the correction method. The use of a light box for collecting color data is essential to have constant lighting conditions and avoid errors in the measurements.

As shown in the article, the correction method proposed is a very suitable approach for the color correction of measurements taken with different mobile devices. In every experiment, the correction method standardizes each measurement to a common and well-defined system, the sRGB color system, but it also significantly reduces the relative errors of the measurements and the variance between smartphones, making the measurements comparable. As color can be related to the concentration of many different compounds, many different quantitative methods using smartphones can be considered.

Despite being such a useful methodology, it presents certain limitations that must be improved. For better quantitative results, other types of illuminants will have to be considered, so the initial lighting conditions are more similar to the reference color system. Regarding the Light Box, it should be redesigned to improve the light distribution among the samples, reducing the errors associated with light gradients or the glare produced on the surface of the sample. Currently, a new light box and a mobile application are being developed by our group so that this methodology can be implemented in an easier and more intuitive way. Our goal is to make analytical color measurements with DSCs more accessible to all users, hoping that in the future, these measurements can be compared with those of, for example, a spectrophotometer.

## Figures and Tables

**Figure 1 biosensors-12-00341-f001:**
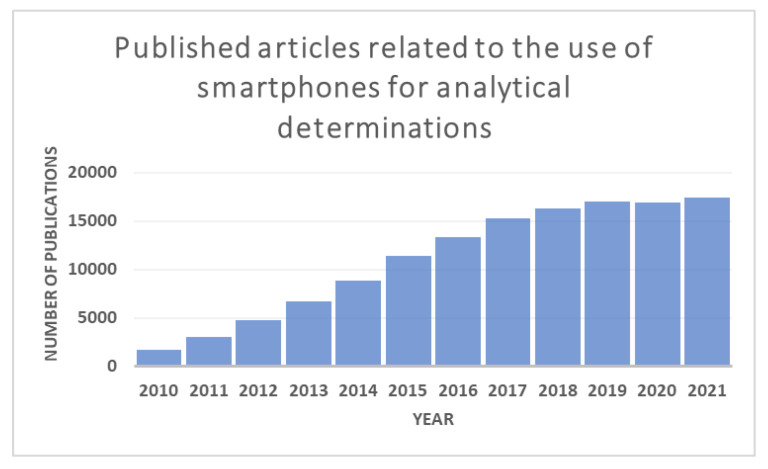
The self-developed graph shows the number of publications related to the use of smartphones for analytical purposes by year. The data was compiled by searching the keywords “Smartphone analytical determinations” in the Google Scholar Research Gate. The graph shows a significant increase in the published articles in the last twelve years.

**Figure 2 biosensors-12-00341-f002:**
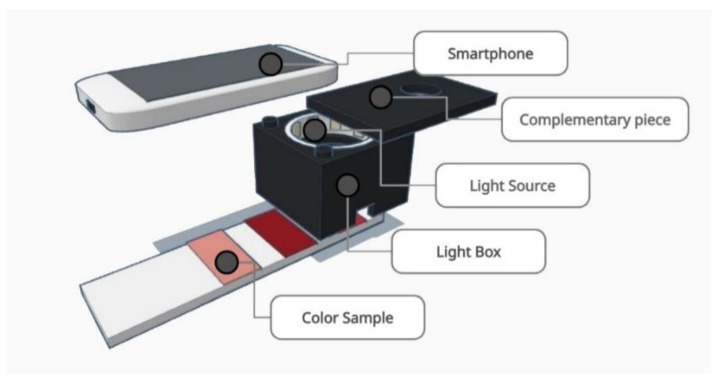
Scheme of the color measurement system used for this article. First, the LED strip is accommodated inside the light box. Second, the complementary piece is mounted on the upper part of the light box. Finally, the color sample is placed behind the light box and the photography is taken with the smartphone.

**Figure 3 biosensors-12-00341-f003:**
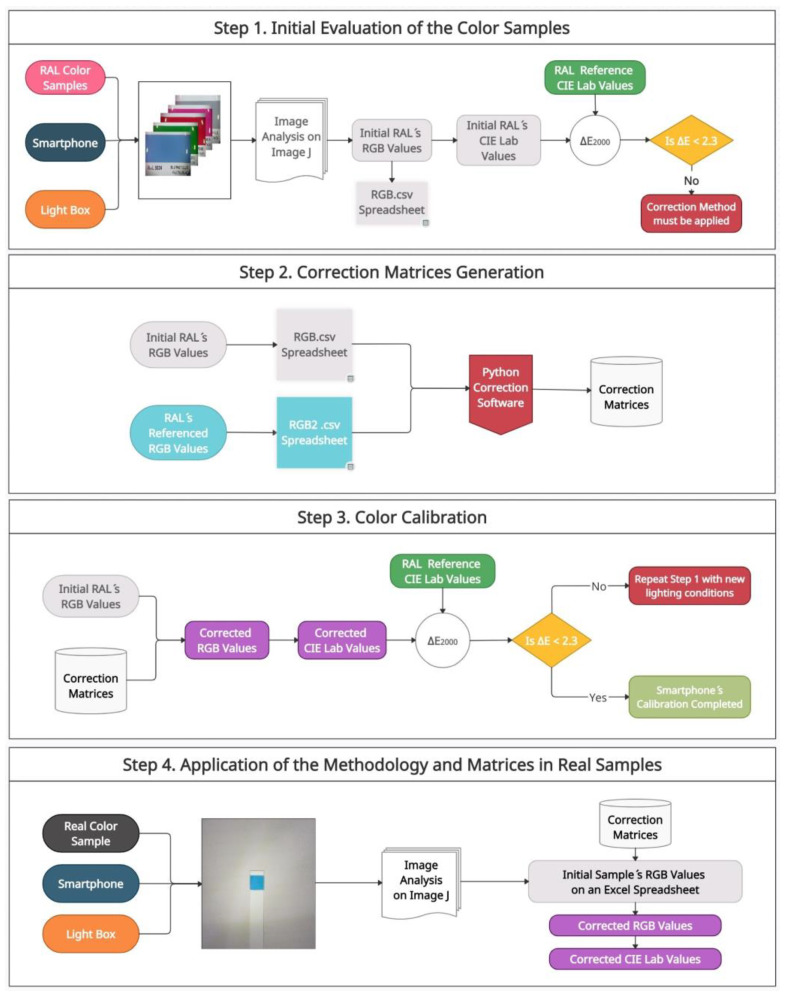
Flowchart for the correction of color samples. The methodology is based on a four-step process: (1) Initial evaluation of the color samples using CIE∆E2000, (2) Correction matrices generation with the Python-based software, (3) Color calibration by applying the matrices generated to the color samples, (4) Application of the methodology and matrices in real samples.

**Figure 4 biosensors-12-00341-f004:**
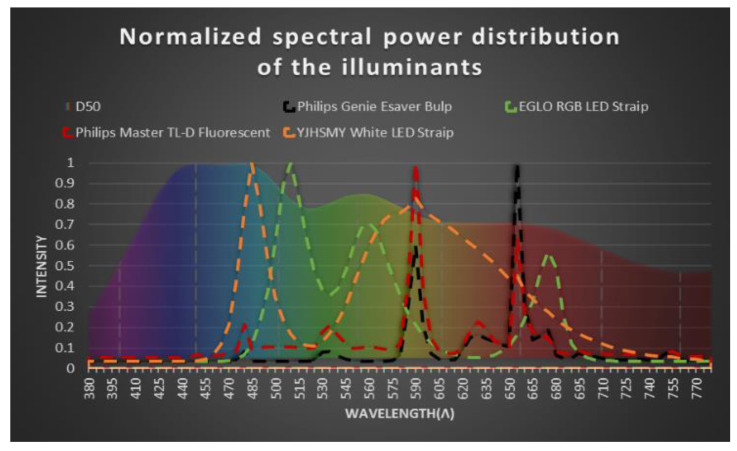
Normalized spectral power distribution of four different illuminants with respect to their maximum wavelength. The individual spectral power distribution of the illuminants is compared with the D_50_ Illuminant represented at the back of the graphic.

**Figure 5 biosensors-12-00341-f005:**
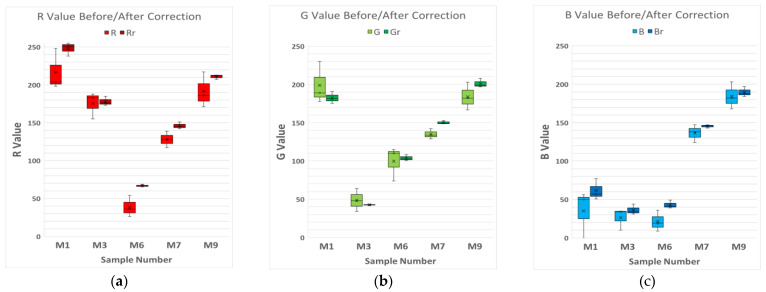
Box and Whisker plot of 5 RAL samples chosen randomly, associated with their RGB values. The lighter colors are the RGB values before the correction and the darker ones are after the correction. (**a**) Red channel values of five samples before/after the correction. (**b**) Green channel values of five samples before/after the correction. (**c**) Blue channel values of five samples before/after the correction.

**Figure 6 biosensors-12-00341-f006:**
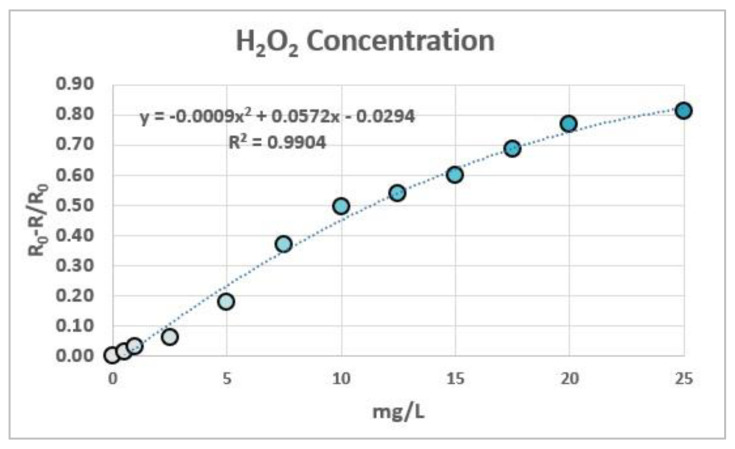
Calibration of 12 solutions of different concentrations of peroxide hydrogen. The plot was adjusted to a second-degree polynomial equation.

**Figure 7 biosensors-12-00341-f007:**
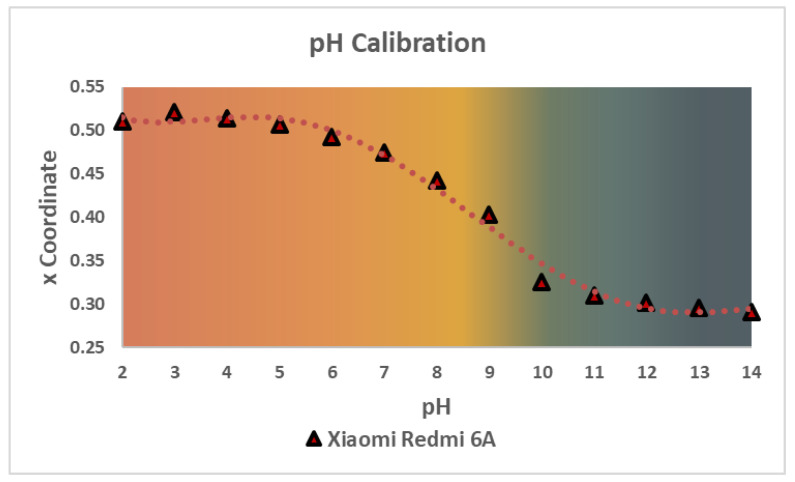
pH calibration of 12 solutions in the range of 2 to 14 in pH. This plot shows the color transition for the PanReac AppliChem^®^pH strips between the pH values measured with the Xiaomi Redmi 6A Smartphone.

**Figure 8 biosensors-12-00341-f008:**
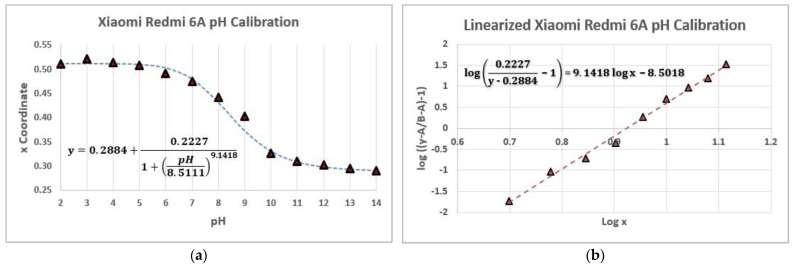
pH measurements with a Xiaomi Redmi 6A Smartphone. (**a**) Logistic function of the color measurements. (**b**) Linearized form of the logistic function for the color measurements. For the best-linearized fit, the values of pH 2, 3, 4, and 14 were discarded.

**Table 1 biosensors-12-00341-t001:** Evaluation of 55 RAL color samples with the CIE∆E_2000_ (2:1:1) criterion, CI (95%).

**Without Correction**	**∆L**	**∆a**	**∆b**	**∆E_2000_**
Mean	7.82 ± 0.81	11.27 ± 1.51	8.18 ± 1.43	6.88 ± 0.45
**With Correction**	**∆L**	**∆a**	**∆b**	**∆E_2000_**
Mean	1.31 ± 0.29	1.99 ± 0.36	2.25 ± 0.39	1.95 ± 0.24

**Table 2 biosensors-12-00341-t002:** Evaluation of 55 RAL color samples with the CIE∆E_2000_ (2:1:1) criterion using different illuminants, CI (95%).

**Illuminant**	**Without Correction**	**∆L**	**∆a**	**∆b**	**∆E_2000_**
EGLO RGB LED Strip	Mean	10.93 ± 1.38	5.03 ± 1.20	24.48 ± 2.94	15.87 ± 2.19
Philips Genie Esaver Bulp		5.34 ± 1.13	10.10 ± 2.02	27.76 ± 3.24	13.30 ± 1.35
Philips Master TL-D Fluorescent		6.08 ± 1.11	4.63 ± 0.83	14.00 ± 1.58	8.33 ± 0.78
YJHSMY White LED Strip		5.83 ± 0.99	9.97 ± 1.66	9.77 ± 1.50	6.65 ± 0.58
**Illuminant**	**With Correction**	**∆L**	**∆a**	**∆b**	**∆E_2000_**
EGLO RGB LED Strip	Mean	3.86 ± 0.67	3.56 ± 0.81	3.67 ± 0.76	4.12 ± 0.53
Philips Genie Esaver Bulp		2.29 ± 0.43	2.47 ± 0.63	5.32 ± 1.04	3.76 ± 0.56
Philips Master TL-D Fluorescent		2.90 ± 0.58	2.79 ± 0.77	3.02 ± 0.75	2.98 ± 0.45
YJHSMY White LED Strip		2.35 ± 0.46	2.44 ± 0.58	2.63 ± 0.54	2.61 ± 0.27

**Table 3 biosensors-12-00341-t003:** Standards deviation mean of the RGB channels between devices that have controlled and non-controlled lighting conditions (*n* = 55).

Device	Lighting Conditions	R_SD_	G_SD_	B_SD_
Xiaomi Redmi 6A	Non-controlled	3.68	3.75	3.93
HP Scanjet G2410	Controlled	1.91	1.89	1.94

**Table 4 biosensors-12-00341-t004:** Standards deviation mean of the RGB channels of a Xiaomi Redmi 6A Smartphone using/not using a light box, (*n* = 55).

Device	Lighting Conditions	R_SD_	G_SD_	B_SD_
Xiaomi Redmi 6A	Non-controlled	3.68	3.75	3.93
	Controlled	1.17	1.10	1.33

**Table 5 biosensors-12-00341-t005:** Evaluation of 55 RAL color samples with the CIE∆E_2000_ (2:1:1) criterion using/not using a Light Box, CI (95%).

**Without Correction**	**∆L**	**∆a**	**∆b**	**∆E_2000_**
Without Light Box	5.83 ± 0.99	9.97 ± 1.66	9.77 ± 1.50	6.65 ± 0.58
With Light Box	5.52 ± 0.98	8.01 ± 1.44	7.25 ± 1.38	5.82 ± 0.46
**With Correction**	**∆L**	**∆a**	**∆b**	**∆E_2000_**
Without Light Box	2.35 ± 0.46	2.44 ± 0.58	2.63 ± 0.54	2.61 ± 0.27
With Light Box	1.90 ± 0.38	2.05 ± 0.46	1.93 ± 0.40	2.01 ± 0.22

**Table 6 biosensors-12-00341-t006:** Evaluation of 10 color samples with three different smartphones. Comparison and matching on the RAL Chart with correction (WC) and without correction (WOC).

Smartphone	Nº Sample	Reference	WOC	Match	WC	Match
Xiaomi Redmi 6A	1	RAL 1018	RAL 1016	No	RAL 1032	No
	2	RAL 2008	RAL 2011	No	RAL 2008	Yes
	3	RAL 3016	RAL 3020	No	RAL 3020	No
	4	RAL 4011	RAL 4005	No	RAL 4011	Yes
	5	RAL 5007	RAL 5002	No	RAL 5023	No
	6	RAL 6010	RAL 6037	No	RAL 6010	Yes
	7	RAL 7001	RAL 7033	No	RAL 7030	No
	8	RAL 8025	RAL 8028	No	RAL 8025	Yes
	9	RAL 9010	RAL 9002	No	RAL 9010	Yes
	10	RAL 9017	RAL 9005	No	RAL 9005	No
Xiaomi Redmi 4A	1	RAL 1018	RAL 1016	No	RAL 1032	No
	2	RAL 2008	RAL 2011	No	RAL 2008	Yes
	3	RAL 3016	RAL 3013	No	RAL 3000	No
	4	RAL 4011	RAL 4011	Yes	RAL 4011	Yes
	5	RAL 5007	RAL 5000	No	RAL 5023	No
	6	RAL 6010	RAL 6002	No	RAL 6010	Yes
	7	RAL 7001	RAL 7046	No	RAL 7042	No
	8	RAL 8025	RAL 8028	No	RAL 8025	Yes
	9	RAL 9010	RAL 7047	No	RAL 9001	No
	10	RAL 9017	RAL 9005	No	RAL 9011	No
iPhone SE	1	RAL 1018	RAL 1016	No	RAL 1032	No
	2	RAL 2008	RAL 1033	No	RAL 2008	Yes
	3	RAL 3016	RAL 2002	No	RAL 3000	No
	4	RAL 4011	RAL 4008	No	RAL 4011	Yes
	5	RAL 5007	RAL 5002	No	RAL 5007	Yes
	6	RAL 6010	RAL 6017	No	RAL 6010	Yes
	7	RAL 7001	RAL 7045	No	RAL 7042	No
	8	RAL 8025	RAL 8025	Yes	RAL 8025	Yes
	9	RAL 9010	RAL 9001	No	RAL 9002	No
	10	RAL 9017	RAL 9005	No	RAL 9005	No

**Table 7 biosensors-12-00341-t007:** Evaluation of five solutions of different concentrations of H_2_O_2_ (*n* = 3) with three different devices with correction (WC) and without correction (WOC).

Smartphone	Nº Sample	Real [H_2_O_2_] (mg/L)	WOC (mg/L)	WC (mg/L)	Relative Error WOC (%)	Relative Error WC (%)
Xiaomi Redmi 6A	1	1.36	3.60 ± 0.19	1.69 ± 0.13	164.89	24.08
	2	7.82	11.47 ± 0.36	8.88 ± 0.30	46.79	13.56
	3	13.6	21.22 ± 0.72	16.81 ± 0.59	56.06	23.65
	4	20.4	N/A	21.29 ± 0.47	N/A	4.37
Xiaomi Mi A2	1	1.36	2.35 ± 0.06	1.95 ± 0.07	73.27	43.47
	2	7.82	9.30 ± 0.27	8.92 ± 0.42	19.02	14.09
	3	13.6	18.00 ± 0.34	16.96 ± 0.17	32.38	24.76
	4	20.4	N/A	21.32 ± 0.47	N/A	4.54
iPhone 8	1	1.36	2.53 ± 0.13	1.75 ± 0.15	86.28	29.06
	2	7.82	9.91 ± 0.21	8.84 ± 0.18	26.75	13.11
	3	13.6	18.98 ± 0.32	16.97 ± 0.20	39.56	24.77
	4	20.4	25.54 ± 1.10	21.21 ± 0.53	25.19	3.99

**Table 8 biosensors-12-00341-t008:** RGB Values of the Xiaomi Redmi 6A measurements. Range of measurement: pH 2–14.

pH	Color	Corrected RGB (Xiaomi Redmi 6A)
2		204,93,68
3		(230,118,57) ^1^
4		(238,124,58) ^1^
5		(247,134,61) ^1^
6		(249,149,65) ^1^
7		(246,162,68) ^1^
8		(202,157,64) ^1^
9		152,134,71
10		83,105,70
11		76,100,76
12		74,95,80
13		65,85,75
14		57,67,68

^1^ These measurements had a value of 0 in the blue channel before the correction so they have an initial error associated.

**Table 9 biosensors-12-00341-t009:** Evaluation of four sample buffers (*n* = 3) of different pH with three different devices, the data are shown with and without the correction method applied (WC, WOC), CI 95%.

Smartphone	Nº Sample	Real pH	WOC pH	WC pH	Relative Error WOC (%)	Relative Error WC (%)
Xiaomi Redmi 6A	1	7.35	6.62 ± 0.23	7.28 ± 0.11	9.89	0.84
	2	8.74	8.98 ± 0.03	8.76 ± 0.05	2.81	0.24
	3	9.83	9.99 ± 0.22	9.85 ± 0.14	1.70	0.30
	4	5.11	N/A	5.87 ± 0.24	N/A	14.94
Xiaomi Redmi 5 Plus	1	7.35	8.07 ± 0.02	7.27 ± 0.03	9.87	1.09
	2	8.74	9.15 ± 0.04	8.91 ± 0.04	4.76	2.06
	3	9.83	9.91 ± 0.15	9.80 ± 0.10	0.85	0.32
	4	5.11	7.40 ± 0.02	5.63 ± 0.33	44.80	10.17
Huawei P30 Lite	1	7.35	7.14 ± 0.08	7.23 ± 0.06	2.77	1.57
	2	8.74	9.03 ± 0.14	8.93 ± 0.14	3.36	2.26
	3	9.83	9.60 ± 0.04	9.83 ± 0.06	2.25	0.03
	4	5.11	N/A	5.91 ± 0.12	N/A	15.79

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
