# Peer review of "Solving Color Reproducibility between Digital Devices: A Robust Approach of Smartphones Color Management for Chemical (Bio)Sensors"

_biosensors, 2022, doi:10.3390/bios12050341_

Round 1

Reviewer 1 Report

1. The objective of the work should be highlighted in the introduction section.
2. For the developed method, do you consider the computation burden issue? For this case, i think you need to discuss more after the main results and a remark will be helpful here. What the difficulties you have met when deriving the current results? The authors are suggested to add a remark after the main results.
3. In the experimental part, the detail parameters used in the proposed methodology are not given.
4. The authors have hardly taken effort to discuss the proposed work. This demotivates the reader. It should be more clearly discussed.
5.  The results are not clearly explained in alignment to the objective of the paper.
6.  Some standard metrics are used by the authors to measure the performance of the proposed method. What are the standard acceptable values of performance metrics? Why authors choose performance metrics? Must explain the reasons.
7. Abbreviations throughout text need to be revised. In some instances, abbreviations are mentioned without the original phrase.
8. The computational complexity either in terms of calculations required or in terms of execution time may also be computed and compared.

Minor Comments:
*In Section 1, it is better Section Contributions follows Section Related Works. The authors did not clearly investigate the relate works. It is better the authors investigate all the related works and then state their contributions.
* The contributions and the hypothesis should be presented clearly in a concise way,
* To have an unbiased view in the manuscript, there should be some discussions on the limitations of the proposed method. Additionally, clearly justify the novelty and innovative insights of this manuscript,
* Revisions in terms of detailed experiments and discussions can be re-checked,
* Discussions of the experimental results achieved can be enhanced, being objective and concise,
* The innovation of the strategy/method is not clear described enough, need to provide more details and explain further,
* The linguistic quality needs improvement. It is essential to make sure that the manuscript reads smoothly- this definitely helps the reader fully appreciate your research findings. 
* Add industrial significance of the proposed approach,

if input data are noisy, How your method works on noisy data? Make a remark. Some latest related work should be included in literature.

Reviewer 2 Report

Paper : « Solving color reproducibility between digital devices: A robust approach of Smartphones color management for chemical (bio)sensors »

General advice : The present paper deals with a real problem. The authors have realized a substantial and serious work, very useful for readers interested in color interpretation. Nevertheless, the paper is mainly based on existing tools and softwares and does not propose novel notions or formulas. Strictly speaking, it does not correspond to a research work and can be considered as a high-tech development.

The subject is clearly exposed and the figures are useful and well commented : as an example, let us cite Fig. 3, presenting the four steps necessary to evaluate the DSC´s initial RGB values and apply the correction method.

The presentation of some technical subjects as « supplementary materials » is a good idea, that makes reading more fluid.

Some remarks and questions

The objective of the paper requires the use of tools taking into account the Human Visual System. This is the reason why it is appropriate to consider the CIE Lab System and the distance ΔE. However, in section 2.7.2. (Correction matrices generation), the authors use an existing Python code based on a linear fit using least squares in the RGB space. Then in section 2.7.3. (Color calibration) the corrected initial color values are converted to the CIE Lab system and the ∆E values are recalculated. My question is : is it true that the minimization performed in the RGB space corresponds to a minimization with the ΔE distance ? If not, the future cases where ΔE is greater than the acceptable threshold of 2.3 could be generated by the spaces changing.

Other points need more explanation in my opinion. In section 3.1., it is said that 15 of the 96 initial color samples did not meet the criteria of JND (∆E < 2.3), as it is shown in the Supplementary Material 8, so they were discarded, having 81 color samples for the next study. My question is : how do you justify the elimination of « bad » samples ?

In Table 1, the presented results are « given by means of the average values of L, a, b and ∆E2000 belonging to the 81 RAL ». When you want to compare two distributions (for example ∆L values before and after correction, considering average values constitutes a very simple tool. Moreover, concerning the comment after Table 1, my remark is : The decreasing of ∆E values is an interesting result showing that the method performs some correction, but it does not prove that « the sensor and lighting measurement conditions can be corrected and referenced to a unique color system ».  

Lines 254-258 : The elimination of color samples whose R, G or B coordinates were equal to 0 or 255 could be done from the beginning.

Table 3 : The correction method is proved to be efficient to balance the effects of various illuminants, but the corresponding (corrected) ∆E are greater than 2.3. This point needs a comment from the authors. different from « the correction method can be applied to any kind of illuminants » (line 298).

Table 7 is not easy to interpret and needs more explanations. Please describe precisely the content of each column (RAL N°, meaning of Match/Mismatch…).

The practical application (section 3.6) concerns the determination of H2O2 of various solutions. Three devices are tested with five different concentrations varying from 0.6 mg/l to 20.4 mg/l. Table 8 presents the estimated concentrations with and without correction. Fortunately, the error percentages are lower after correction. Nevertheless, it appears that the observed errors are much more important for low concentrations, until 93.28% !!! Being not a specialist of chemical biosensors, I expect from the authors some more information about the real applicative interest of such imprecise estimations.

The second application (section 3.7) concerning the determination of the different pH colors of various solutions appears much more convincing. In fact, the relative errors  presented in Table 10 are very low.

I think the authors must rewrite the paper, explain that the presented work is a first step and try to suggest perspectives to improve the current results.

Round 2

Reviewer 1 Report

The paper is acceptable now. 

Author Response

We have reviewed the English language throughout the paper

Thank you very much.

Reviewer 2 Report

Comments on the paper « Solving color reproducibility between digital devices: A robust approach of Smartphones color management for chemical (bio)sensors » (revised version).

I appreciate the real efforts made by the authors to take into account the remarks and suggestions I proposed in my previous review. It is now much easier for the reader to understand the goals and reasoning developed. Moreover, the results are honestly presented and the future works remaining to perform are clearly mentioned. The suppression of Table 1 contributes to these clarifications.

Concerning the answer to my question: “is it true that the minimization performed in the RGB space corresponds to a minimization with the ΔE distance?”, the authors propose two possible interpretations and the second is the good one: 2) Is the ΔE criteria also adequate when working with RGB values? I confirm that this point would need a complementary study.

At the end of their answers, the authors announce that they are developing « an APP to be implemented on any smartphone to make these corrections automatically on the pictures ». I think this would be very useful and I have a last question/suggestion in link with the comment made by the authors on my remark n° 4 (Table 3): They say: “It is observed that, although in all cases the ∆E are reduced, these values are greater than 2.3, probably due to the fact that there are not controlled lighting conditions". For this APP project, would it be feasible and potentially effective to introduce an image pre-processing to balance possible lighting variations?

 Finally, I consider that the paper can be now accepted for publication.
